# Research on the Mechanical Behavior of Buried Double-Wall Corrugated Pipes

**DOI:** 10.3390/polym14194000

**Published:** 2022-09-24

**Authors:** Dongyang Gao, Huiwei Yang, Wenwen Yu, Xiaogang Wu, Angxuan Wu, Guoyun Lu, Qiang Zheng

**Affiliations:** 1School of Civil Engineering, Taiyuan University of Technology, Taiyuan 030024, China; 2School of Materials Science & Engineering, Taiyuan University of Technology, Taiyuan 030024, China; 3School of Biomedical Engineering, Taiyuan University of Technology, Taiyuan 030024, China

**Keywords:** interior and exterior wall thicknesses, circumferential strain, radial displacement, soil load

## Abstract

The mechanical behavior of buried HDPE double-wall corrugated pipes is mainly affected by the material and the structure of the pipe wall. Here we studied a peculiar material that added fly ash (FA) in high density polyethylene (HDPE) to develop composites. We have conducted research on FA/HDPE composites with different mix proportions. When 5% compatibilizer was added to the 10% FA masterbatch/HDPE composite, the Young’s Modulus of FA/HDPE composite was higher. This paper mainly studies the mechanical behavior of the structure of pipe walls for materials with this proportion of the ingredients. The mechanical behavior of double-wall corrugated pipes with different ratios of interior and exterior wall thicknesses is studied by keeping the sum of the interior and exterior wall thicknesses unchanged. Pipes with six different ratios of interior and exterior wall thicknesses are simulated; the results show that the strain of crest and liner gradually decreased and the valley strain gradually increased with the increase of the exterior wall thickness. By comparing inner and outer wall thickness ratios from 0.67 to 2.33, it is found that the structural performance and economic advantage for the double-wall corrugated pipes is best when the thickness ratio of the interior wall and the exterior wall is controlled to be from 1.3 to 1.8. This paper expounds the deformation mechanism of double-wall corrugated pipes from the perspective of mechanical behavior and structural characteristics, and provides a reference for material selection and structural design of double-wall corrugated pipes.

## 1. Introduction

HDPE double-wall corrugated pipe is a flexible pipe, which has the characteristics of stable chemical properties, light weight, low temperature resistance and excellent mechanical properties. The double-wall corrugated pipes have a special structure of pipe wall, consisting of a periodically distributed corrugated exterior wall and a smooth liner wall. The valley of the double-wall corrugated pipes is located in the middle of the two corrugations, and the wall thickness of the valley is the sum of the exterior wall and the interior wall. This structural distribution makes the ring stiffness of the double-wall corrugated pipes better than the straight-walled pipes and corrugated pipes of the same wall thickness. The interface characteristics of the double-wall corrugated pipes are shown in Figure 1.

With the development of materials science, the advantages of composite materials in tensile strength, elongation at break, etc. have gradually attracted attention. Zaghloul et al. carried out studies on the mechanical properties of nano-reinforced particle polyester matrix composites and glass fibre-reinforced polyester [1,2], and Zaghloul et al. have also carried out studies on the effect of different filler materials on the mechanical properties of polymers [3,4]. Xu, L., Jin, T. and others conducted quasi-static and dynamic compression tests on polyethylene materials, and the tangential moduli of the stress-strain curves at different strain rates were approximately the same [5,6]. The production and processing process of the polymer is an important factor in determining the performance of the polymer. Through the study of the electrophoretic deposition process of polyaniline, Zaghloul et al. proposed that research can predict the thermal stability of the polymer and reduce the thermal degradation of the polymer during application [7,8]. Chinese specifications are considered when conducting tensile tests of polymers [9]. Through the laboratory and FEM, Wu et al. proposed a composite of fly ash/HDPE to feasibly enhance the stiffness of double-wall corrugated pipes [10].

There are two directions for study of the external load of the pipes; one is the ground load including traffic load [11,12], collapse impact load [13], and ground circular loads [14] etc. Another direction is the effect of geology and soil layers, including seismic response [15,16], geotechnical loads [17], soil subsidence [18,19], and soil layer displacement [20,21,22], etc. When the buried depth of the pipes gradually deepens, the influence of the ground load on the pipes becomes smaller, and the influence of the soil gradually increases. Studies have shown that the effect of geological activity on the pipes is affected by the diameter, burial depth, and the bearing capacity of the pipes located at the fault is much lower than that of other locations. The mode of action of the external load can have an effect on the failure of the pipes. Furthermore, Das and Dhar explored the effect of loading rate on the bearing capacity of the pipes [23]. Bilgin and Stewart studied the interface shearing resistance of polyethylene pipes, as the pipe diameter becomes smaller, the interface shear resistance and normal stress both decrease [24]. Hao, W. et al. studied the plastic hinge mechanism of bellows, and summarized the mechanism of energy absorption deformation of bellows [25].

The interaction between pipes and soil is also the focus of pipes research. Chinese specifications are considered when studying backfill properties and vehicle loads [26,27,28,29,30]. The friction coefficient between high-density polyethylene pipes and surrounding soil is affected by the operating temperature of the pipes. The higher the operating temperature of the pipes, the greater the friction coefficient between the pipe and soil [31]. In addition, the change of pipe-soil interaction caused by different soil properties is also an important research direction. The mechanical properties of double-wall corrugated pipes of various diameters were analyzed under the action of surrounding soil with various compactness [32,33]. In finite element analysis, the Mohr–Coulomb model is often used to simulate soil properties. Since the traditional Mohr–Coulomb model cannot simulate unsaturated soil loads accurately, Robert proposed an unsaturated modified Mohr–Coulomb model to capture the realistic loading induced by unsaturated soil medium [34]. Based on experimental and numerical investigation. Moradi and Abbasnejad used a modified Mohr–Coulomb model to study the arching effect over a trapdoor in sand [35]. Through the study of backfill stiffness and pipe-soil interaction, Morteza and Hodjat revealed the influence of different backfill stiffness on pipe-soil interaction and pipe failure model [36].

In summary, researchers in this field have conducted researches on the interaction between pipes and soil [30], and have conducted in-depth analysis of the mechanical response to double-wall corrugated pipes under the action of backfill soils with different degrees of compaction [31,32]. However, the structural form of the double-wall corrugated pipes is affected by factors such as wall thickness, corrugation spacing, corrugation height, corrugation form, so it is complicated to research the structure of the double-wall corrugated pipes. This paper uses ABAQUS software to establish a three-dimensional pipe-soil model to study the mechanical characteristics of different interior and exterior wall structures for FA/HDPE double-wall corrugated pipes. In addition, this paper also focuses on summarizing the change law of mechanical behavior with the change of interior and exterior wall thickness ratio.

## 2. Materials of FA/HDPE Double-Wall Corrugated Pipes

Wu et al. have carried out research on the tensile properties of FA/HDPE materials, and this paper conducts research on practical engineering applications on the basis of materials research [10]. Tensile tests were carried out according to GB/T 1040.2-2006 [9] specification. Five specimens were tested per batch and the average strength was calculated; the specimens were stretched at a rate of 5 mm/min until the specimens failed. The tensile specimen size chart is shown in Figure 2.

The tensile test results of different contents of FA/HDPE are shown in Table 1. When the content of the FA masterbatch is less than 10%, the tensile yield strength and tensile rupture strength of FA/HDPE composites increase with the increase of FA masterbatch content. When the content of the FA masterbatch was greater than 10% and less than 20%, the tensile yield strength and tensile breaking strength changed little. Furthermore, the Young’s modulus of FA/HDPE decreases with the increase of the content of FA masterbatch. Because rigid FA cannot be stretched together with the HDPE, the elongation at the break of FA/HDPE polymer is lower than that of pure HDPE. As the content of FA masterbatch increased in the polymer, the toughening effect of rigid FA gradually became more significant, and the elongation at the break of FA/HDPE gradually increased. The mechanical properties of FA/HDPE can be improved when the content of about 5% of the compatibilizer is added. When the compatibilizer is more than 5%, the mechanical properties of FA/HDPE decrease with the increase of the compatibilizer [10]. The stress-strain diagrams of different composites are shown in Figure 3.

The effect of FA and compatibilizer content in blend composition on Young’s modulus and tensile strength at yield were analyzed [10], as the difference in content can lead to changes in mechanical properties of materials. The pipes made of 10% modified FA masterbatch/5% Compatibilizer/85% HDPE composite could achieve better mechanical properties. The following calculations for FA/HDPE material use the 10% modified FA masterbatch/5% Compatibilizer/85% HDPE composite.

## 3. Finite Element Model

### 3.1. Models of Pipes and Soils

The backfill needs to be backfilled to the top of the pipes in layers and the compaction degree of various soil layers is different. The simulation conditions in this paper are mainly based on practical engineering, considering the practical load effect for buried pipes under the action of multi-layer confining soil.

The backfill process and backfill method of the soil determine the degree of soil compaction. The elastic modulus, Poisson’s ratio and cohesion of the soil are affected by the compaction degree of the backfill. Among these factors, the deformation of the pipes is more sensitive to the change of the elastic modulus of the soil. According to the analysis of finite element simulation and experimental results [32], it is reliable to use the elastic modulus as the control variable of the compaction degree. In this paper, the compaction degree of the backfill soil is studied by changing the elastic modulus. The material properties of the backfill layer are shown in Table 2.

### 3.2. Meshing of Pipes and Soils

As shown in Figure 1, the exterior surface of the double-walled corrugated pipe is a regular corrugated structure, and the soil in contact with the double-walled corrugated pipe also has a regular corrugated structure, which makes it difficult to mesh the soil around the pipe. Therefore, a locally refined meshing method is used for the soil around the pipes. This method can greatly reduce the number of units of the confining soil around the pipes, improve the running speed of the structure calculation, and improve the calculation accuracy of the pipes and surrounding soils, especially the contact soil around the pipes. Elements of soil and pipes are shown in Figure 4 and Figure 5.

### 3.3. Calculation Conditions for Simulation

The numerical models are specified as follows:(1)It is assumed that the pipes and soil are always in the elastic deformation during the stress stage. Through the study of semi-crystalline polymer yield behavior and macroscopic phenomena, Jin Tao found that although the semi-crystalline polymer material produces nonlinear changes at the end of the elastic stage [6], a linear elastic model can still be used to simulate the elastic stage of the semi-crystalline polymer. Under quasi-static conditions of polyethylene materials, when the stress and deformation of the material is within the yield point, the linear elastic model agrees well with the experimental results [5]. The maximum strains calculated in this paper are smaller than the yield point of the material, so the actual elastic strain value of the pipe does not reach the limit of ultimate elastic strain value. At this deformation stage, the deformation of the pipeline is always in an approximately linear elastic stage, so it is credible to use the linear elastic model to simulate the pipe deformation in this paper.(2)The sum of the thickness of the interior and exterior walls of the pipe is 5 mm, and is equal to the thickness of the valley between the corrugations. The production cost of double-wall corrugated pipes is affected by the distribution of interior and exterior wall thicknesses, and proper ratio of interior and exterior wall thicknesses can maximize structural performance and reduce costs. The strain and deformation characteristics of the pipe are studied under different wall thickness ratios by keeping the sum of wall thickness unchanged and adjusting the ratio of the interior and exterior wall thicknesses. The geometric parameters of the pipes are shown in Table 3.

(3)Consider the friction between the pipe and soil, and that the friction coefficient is 0.4. According to specification [28], the friction coefficient between the high-density polyethylene pipes and the medium sand layer ranges from 0.2 to 0.4. According to the research of Wang, Fei and others [31], the friction coefficient between the buried high density polyethylene pipe and the surrounding soil is affected by the working temperature of the pipe. The friction coefficient between the pipes and soil increases with the increase of the difference between the working temperature and the installation temperature. When the working temperature is too high, the friction coefficient between the pipes and soil is even greater than 0.4. This paper comprehensive consideration of pipes’ work environment, the friction coefficient between the double-wall corrugated pipes and the confining soil layer is considered to be 0.4 in this paper.(4)Gravity and ground load are considered in load calculations, and ground load is set to 0.4 Mpa. The boundaries of the pipe-soil model are constrained as follows: on the side and bottom of the pipe, the soil is fixed completely. The upper surface of the soil does not impose any constraints. Both the front and rear sides of the pipe are perpendicular to the bottom surface, only the normal direction are constrained. Based on the static equivalent principle, the road load is transformed into a 40 kPa uniform load applied directly above the soil, acting on the foundation soil at a position of 3.28 m × 1.5 m. The three-dimension model of the pipe and surrounding soils is shown in Figure 6.

(5)The detection section of pipes can be seen from Figure 7. According to its deformation characteristics, this paper selects crest, valley, and liner of the pipe as monitoring points, and compares and analyzes its displacement and strain under the action of the surrounding soil. Based on the structural characteristics of the double-wall corrugated pipes, this paper studies the strain and deformation of the double-wall corrugated pipes about various interior and exterior wall thickness structures. The mechanical characteristics of double-wall corrugated pipes with various interior and exterior wall thicknesses are summarized. These results can provide some reference for the structural design of double-wall corrugated pipes.

## 4. In-Situ Measurement

The experimental process and results of reference [33], including its in-situ measurement, are referred in this paper. The 6 m long HDPE double-wall corrugated pipe is placed in a trench with 6 m length, 2.2 m width and 2.4 m depth. The two sides of the pipes are constrained by the inspection well. Backfill is compacted by a plate vibrator. The underlying soil layer of the pipe is composed of sandy soil with a thickness of 200 mm and the compaction degree of 90%, which covers the undisturbed soil. Backfill with 95% compaction from the bottom of the pipe to the waist provides strong support for the pipes. From the waist to the top of the pipes, the sand body is backfilled around the pipes with 100 mm hoist and compacted to 95% before subsequent backfilling. The soil layer at the top of the pipe is divided into two parts, and the compaction degrees of the filling sand bodies are 85% and 90% respectively. Backfill in-situ soils with 90% compactness from ground to pipe top is 0.4 m. The compactness of each soil layer is shown in Figure 8.

A single-axle and two-wheel truck is selected as the experimental vehicle. The standard axle load of the vehicle is 100 kN, the distance between the wheels of the vehicle is 1.8 m, and the standard load (*p*) is 0.7 MPa. There is a circular load with a diameter of 1.8 m when distributed to the bottom of the groove. The position of the truck tires relative to the pipe is shown in Figure 9. Vertical stress due to vehicle load is related to the thickness of the foundation and the pressure diffusion angle of the foundation [33], and the formula is as follows:(1)$$\sigma_{z1}=\frac{D^{2}p}{{\left(D+2ztan\left(\theta \right)\right)}^{2}}$$
(2)$$\sigma_{z2}={ϒ}_{1}h_{1}+{ϒ}_{2}h_{2}+{ϒ}_{3}h_{3}$$

*D* = diameter of circular load

*p* = ground load

*z* = the distance from the bottom of the groove to the ground

*θ* = foundation pressure spread angle

*h* = thickness of soil layer

## 5. Verification of Number Models

The simulation situation of the pipes is referred to the experimental case, and the reliability of the finite element analysis in this paper is verified by the in-situ experimental results of the reference [33]. The local cylindrical coordinate of the pipe is established in this paper, with the circumferential strain along the direction of the second axis and the radial displacement along the direction of the first axis. The circumferential detection direction and position of the pipe are shown in the Figure 10. The circumferential strain results of numerical simulation and in-situ measurement are shown in Figure 11. The circumferential strain at different positions of the M1 pipe is basically consistent with the in-situ measurement results in terms of numerical value and distribution trend. It shows that the finite element simulation has been fully verified by the in-situ measurement, which proves that the pipe-soil model of the finite element simulation is reasonable and reliable.

## 6. FA/HDPE Double-Wall Corrugated Pipes

The material properties of HDPE and FA/HDPE in the finite element calculation are shown in Table 4. The Poisson ratio of HDPE is greater than that of FA/HDPE, and the density and Young’s modulus are both smaller than FA/HDPE. The comparison of strain between HDPE and FA/HDPE materials of double-wall corrugated pipes has been shown in Figure 12; both pipes use the wall thickness size of M1. Due to the higher Young’s modulus of FA/HDPE material, the circumferential strain is smaller in crest, valley and liner. For the maximum strain position, which located in the valley of the pipes’ sides, the strain is significantly smaller for FA/HDPE pipes than HDPE pipes and the maximum value of the valley is reduced by 29.5%. This effect is conducive to improving the structural performance of the pipes, and improving the safety and durability of the pipes during use.

The comparison of radial displacement between HDPE and FA/HDPE of double-wall corrugated pipes is shown in Figure 13. Due to the higher Young’s modulus for FA/HDPE material, the radial displacement of the FA/HDPE double-wall corrugated pipe is smaller than HDPE double-wall corrugated pipe in the crest and valley. The liner change is smaller than for crest and valley, and the liner displacement of FA/HDPE double-wall corrugated pipes is bigger than for HDPE double-wall corrugated pipes at the top of pipe.

## 7. Analysis of Numerical Simulations

### 7.1. Radial Displacement of Various Interior and Exterior Wall Thicknesses

In order to research the influence of various interior and exterior wall thicknesses of double-wall corrugated pipes on their strain and deformation, a pipe-soil model with the same soil layer parameters and the same pipe diameter was established. The strain and displacement characteristics of various interior and exterior wall thicknesses are analyzed by finite element, and the displacement of the pipes in the radial direction is studied. For double-wall corrugated pipes, the radial displacement of the pipes is usually much larger than the axial displacement. Therefore, this paper takes the radial displacement and circumferential strain of the pipes as the main research objects. Figure 14 and Figure 15 shows the radial displacement of the double-wall corrugated pipes.

It can be seen from Figure 16 that the vertical compression of the double-wall pipes with different interior and exterior wall thicknesses is similar. The difference of interior and exterior wall thickness has little influence on the vertical compression of the pipes. The maximum difference of vertical compression at the crest of the double-wall corrugated pipes is about 3.509%, the maximum difference of vertical compression at the valley is about 3.485%, and the maximum difference of vertical compression at the liner is about 2.765%. As shown in Figure 17, when the surrounding soil is in close contact with the double-wall corrugated pipes, the load outside the pipes directly acts on the valley, causing the pipe wall at the valley to deform toward the inside of the pipes; the rotation angle of the valley and the liner are the same at the junction of the corrugation bottom, so that the displacement direction of the valley is opposite to the liner. Therefore, when the thickness of the exterior wall and interior wall of the double-wall corrugated pipes changes, the vertical compression of the double-wall corrugated pipes in the liner is smaller than the crest and the valley.

### 7.2. Circumferential Strain with Various Interior and Exterior Wall Thicknesses

When the sum of the interior and exterior wall thicknesses remains unchanged, the circumferential strain of different ratios of interior and exterior wall thickness is as shown in Figure 18 and Figure 19. From the overall view of the double-wall corrugated pipes, the strain of the exterior wall and the interior wall increases with the increase of interior and exterior wall thickness ratio. As shown in Figure 20, the maximum strain of the crest occurs on top of the pipes and the minimum strain occurs to the middle area between the two sides of the pipes and the top of the pipes. When the interior and exterior wall thickness ratio increase, the strain at the crest gradually increases, and the maximum strain at the crest always occurs at the top of the pipes. Analyzing the circumferential strain of the valley, whose strain changes at top and bottom of the pipes are larger than other positions, and the maximum strain of the valley decreases with the increase of the ratio of interior and exterior wall thickness. For M6 pipes, the maximum strain of the valley is greater than the maximum strain of the crest. The strain at the liner of the pipes is smaller than crest and valley, because the liner does not directly interact with the surrounding soil, and the liner mainly acts as a connection and constraint between the corrugations. With the change of the interior and exterior wall thickness, the strain change for the liner is significant.

### 7.3. Comprehensive Evaluation of Various Interior and Exterior Wall Thicknesses

As can be seen from Figure 21, when the sum of the interior and exterior wall thicknesses of the double-wall corrugated pipes remains unchanged, reducing the ratio of exterior wall thickness can effectively reduce material cost. By comparing the strain changes of various pipes, reducing the ratio of the interior wall thickness can effectively reduce the local strain at crest and liner, but this method has less effect on the strain of the valley. The maximum strain at the crest and the liner is smaller than the maximum strain at the valley except for M6. Therefore, increasing the thickness ratio of the exterior wall does not solve the problem of local strain on the valley; this makes the local strain of the valley which on the both sides of pipes more significant, so the structure of a thick exterior wall and thin interior wall is unreasonable. For double-wall corrugated pipes whose interior wall thickness is slightly larger than the exterior wall thickness, such as the pipe of M5: although the maximum strain still occurs to the valley of the pipe, the maximum strain at the valley is slightly reduced by increasing the thickness of the interior wall, and the local strain at the valley is alleviated to some extent. Although reducing the thickness of the exterior wall increases the strain and deformation of the crest, it effectively utilizes the bearing capacity of the crest, fully exerts the structural performance of the double-wall corrugated pipe, and reduces the local strain at the valley. By changing the ratio of interior and exterior wall thickness, the double-wall corrugated pipe can redistribute the load, which can effectively improve the bearing capacity and reduce the generation of dangerous sections. When the sum of the interior and exterior wall thicknesses remains unchanged, the change of the interior and exterior wall thickness ratio of the double-wall corrugated pipe within a reasonable range will not cause a significant change in the radial displacement of pipes.

The structure of double-wall corrugated pipes may be considered from the aspects of economy and durability. If the interior wall thickness is slightly larger than the exterior wall thickness, this can achieve better performance in radial displacement, and the economic benefits are higher than those for pipes with thick exterior walls and thin interior walls. Properly increasing the thickness ratio of the interior wall can help to reduce local maximum strain; the maximum strain reduction can effectively improve the safety and durability of the pipes, making the double-wall corrugated pipes more reliable during use.

### 7.4. Comprehensive Indicators of Interior and Exterior Wall Thickness Ratio

As shown in Figure 16, the vertical compression rate of the 6 types of pipes is less than 3%, which meets the engineering technical specification [27]. As shown in Figure 22, the radial displacements are similar for the 6 types of pipes. When the maximum strain of the crest and the valley is close, the structural performance of the double-wall corrugated pipe is fully exerted. According to the above research results, if the structure of the interior wall thickness is slightly larger than the exterior wall thickness, this can make the maximum strain of the crest and the valley close, and can effectively improve the economic benefits. Considering the complex engineering environment and various pipe wall structures, when the ratio of the interior and exterior wall thicknesses of the double-wall corrugated pipe is from 1.3 to 1.8, it has better mechanical properties and economic benefits.

## 8. Conclusions

To discuss the mechanical characteristics of FA/HDPE double-wall corrugated pipes with different interior and exterior wall thickness ratios, the finite element models of pipe-soil interaction were established. Based on comparing the strain distribution, deformation characteristics, production and displacement of different pipes, the following conclusions can be obtained:

(1)When the surrounding soil load acts directly on the wall of the double-wall corrugated pipes, the direction of the depression at the valley is the same as the direction of the force, and the depression direction of the liner is opposite to the valley.(2)FA/HDPE double-wall corrugated pipes have smaller strain and displacement than HDPE double-wall corrugated pipes in the same condition. The maximum strain of FA/HDPE double wall corrugated pipes is about 30% smaller than with HDPE.(3)The strain distribution of double-wall corrugated pipe is significantly affected by the ratio of interior and exterior wall thickness, and different ratios may have different dangerous areas under the action of surrounding soil.(4)When the thickness of the interior and exterior walls changes, the strain changes at the valley are much smaller than the crest and liner.(5)From the perspective of economy and mechanics, when the thickness ratio of the interior wall and the exterior wall is controlled to be from 1.3 to 1.8, this can give a better structural performance for the double-wall corrugated pipes and reduce production cost.

## Figures and Tables

**Figure 1 polymers-14-04000-f001:**
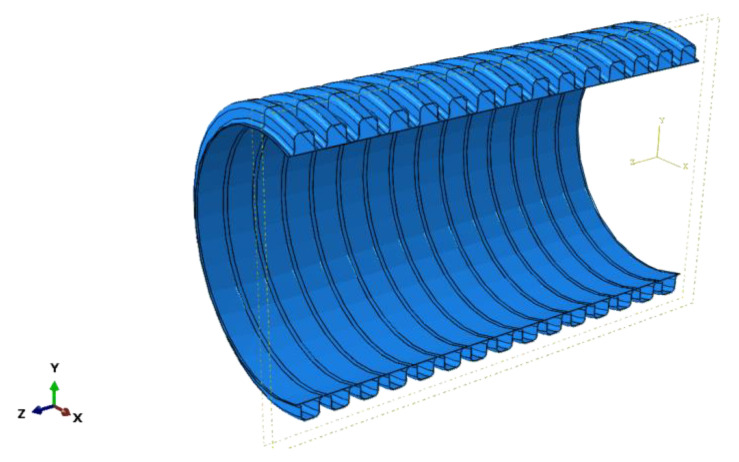
Schematic diagram of cross section of double-wall corrugated pipe.

**Figure 2 polymers-14-04000-f002:**
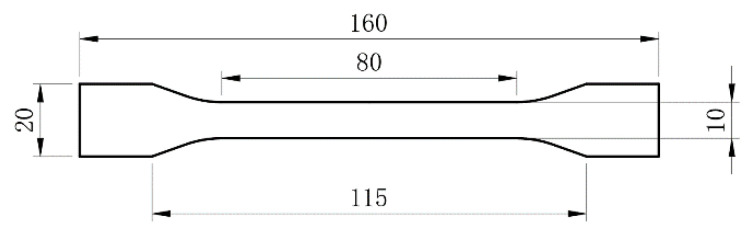
Sample geometry.

**Figure 3 polymers-14-04000-f003:**
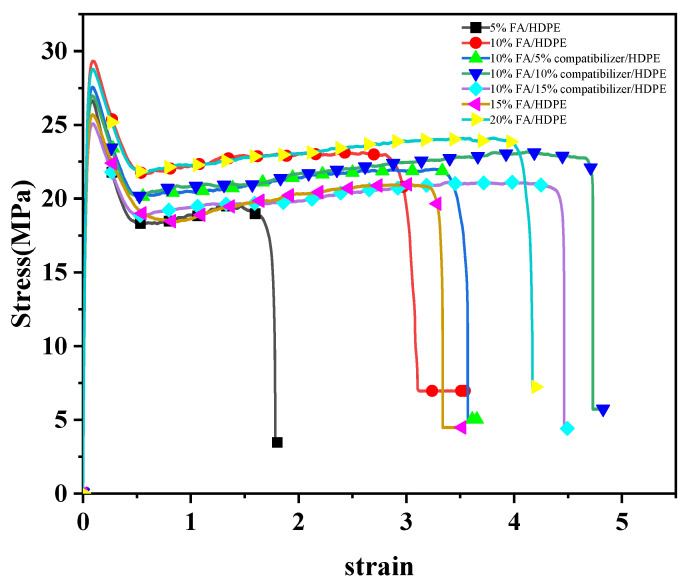
Stress-strain diagrams of different composites [10].

**Figure 4 polymers-14-04000-f004:**
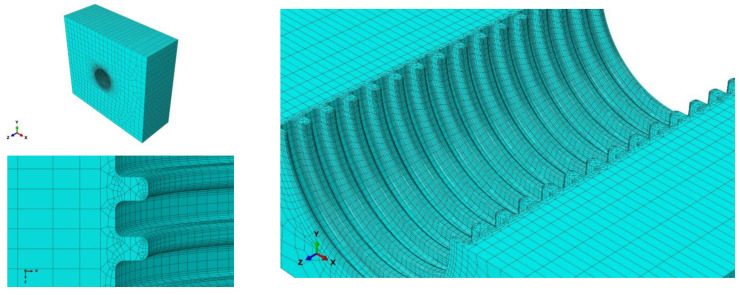
Elements of soil.

**Figure 5 polymers-14-04000-f005:**
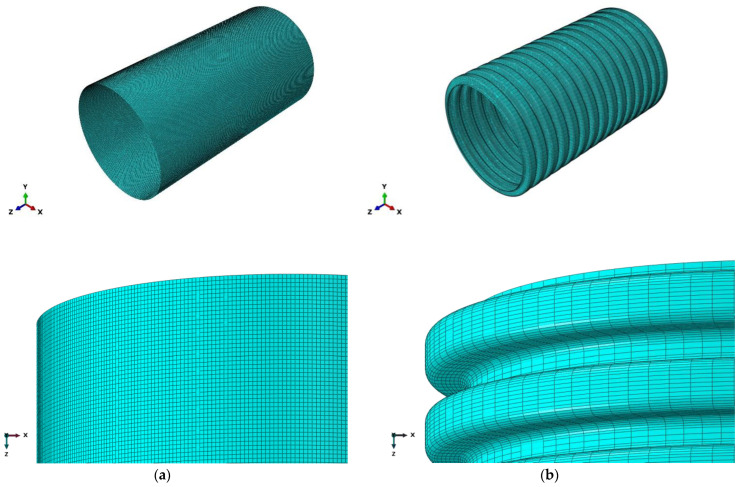
Elements of pipes. (**a**) Interior walls; (**b**) Exterior walls.

**Figure 6 polymers-14-04000-f006:**
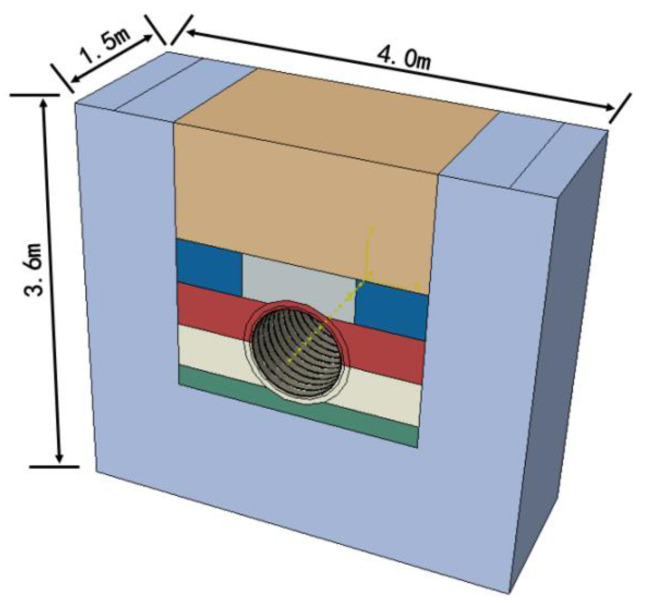
Three-dimension model of the pipe and surrounding soil.

**Figure 7 polymers-14-04000-f007:**
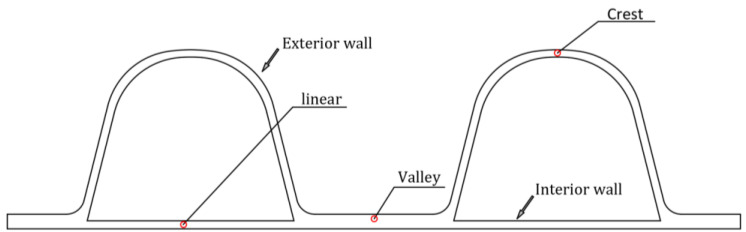
Detection section of pipes.

**Figure 8 polymers-14-04000-f008:**
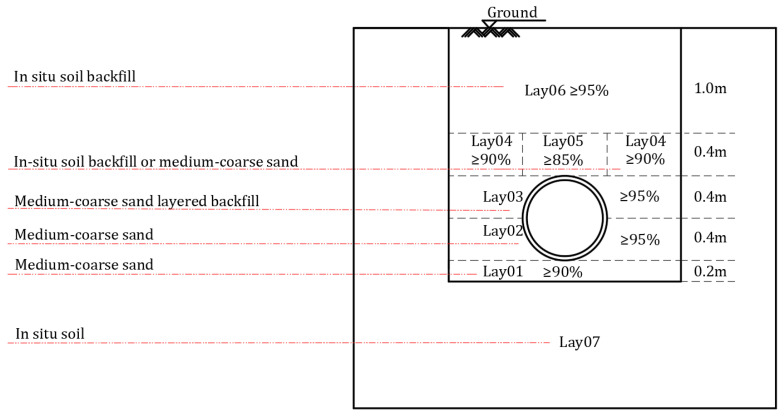
Thickness and compaction of each soil layer.

**Figure 9 polymers-14-04000-f009:**
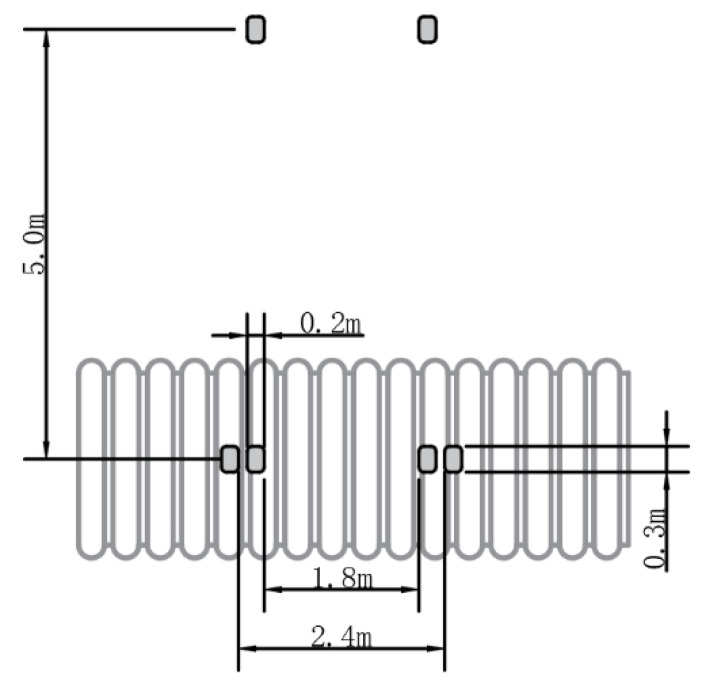
Location and parameters of wheel loads.

**Figure 10 polymers-14-04000-f010:**
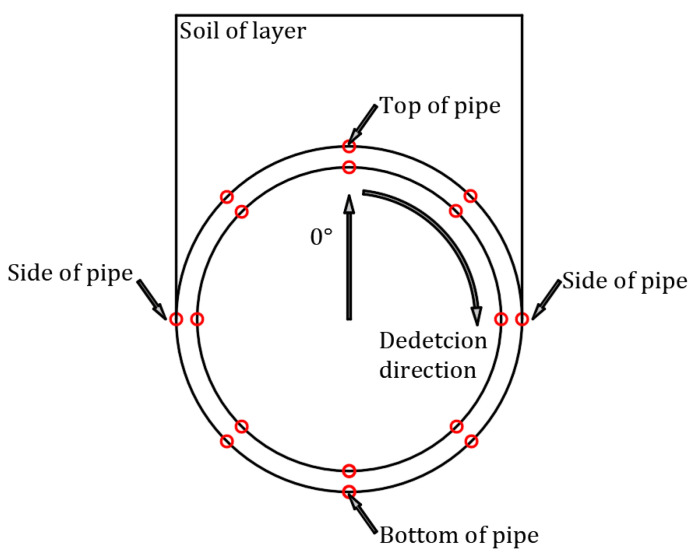
Circumferential detection and position of pipes.

**Figure 11 polymers-14-04000-f011:**
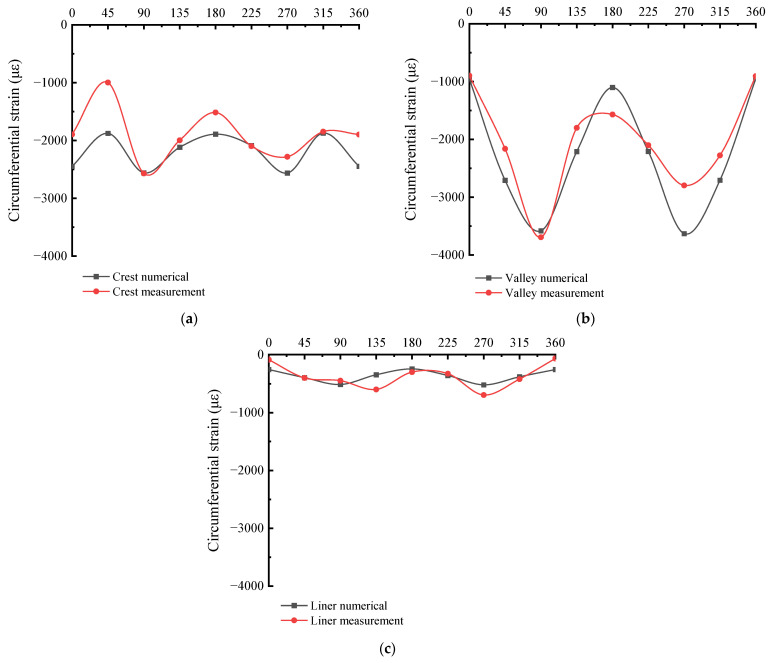
Circumferential strain of pipes. (**a**) Crest; (**b**) Valley; (**c**) Liner.

**Figure 12 polymers-14-04000-f012:**
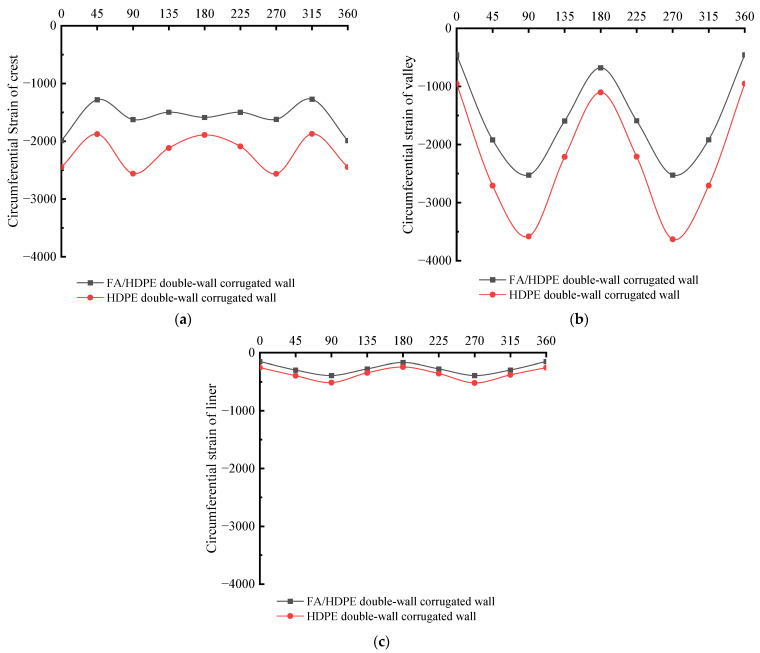
Comparison of circumferential strain between HDPE and FA/HDPE of double-wall corrugated pipes. (**a**) Crest; (**b**) Valley; (**c**) Liner.

**Figure 13 polymers-14-04000-f013:**
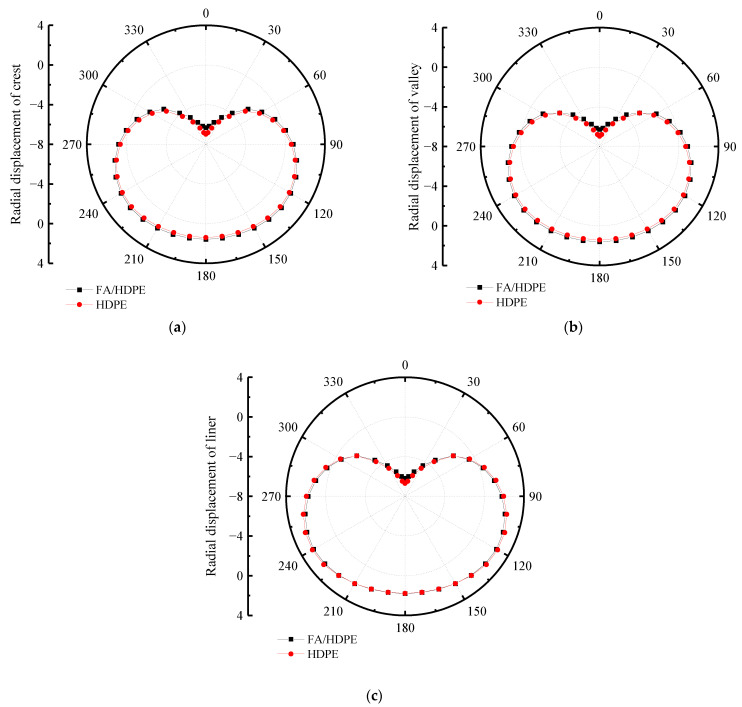
Comparison of radial displacement between HDPE and FA/HDPE of double-wall corrugated pipes. (**a**) Crest; (**b**) Valley; (**c**) Liner.

**Figure 14 polymers-14-04000-f014:**
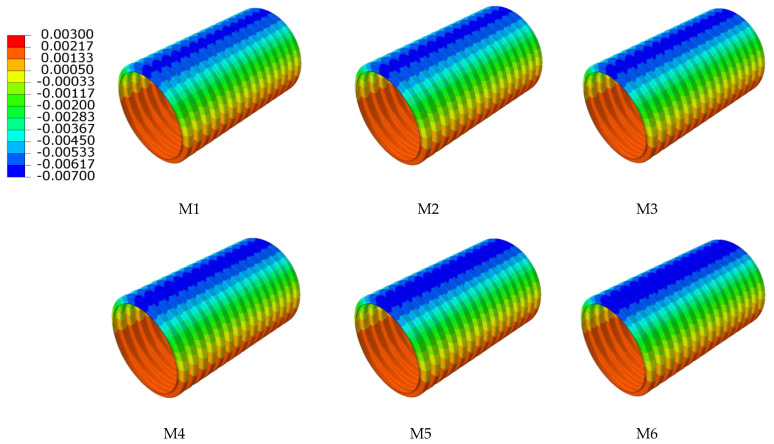
Radial displacement of exterior wall.

**Figure 15 polymers-14-04000-f015:**
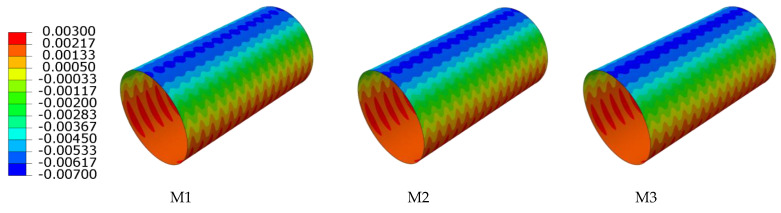
Radial displacement of interior wall.

**Figure 16 polymers-14-04000-f016:**
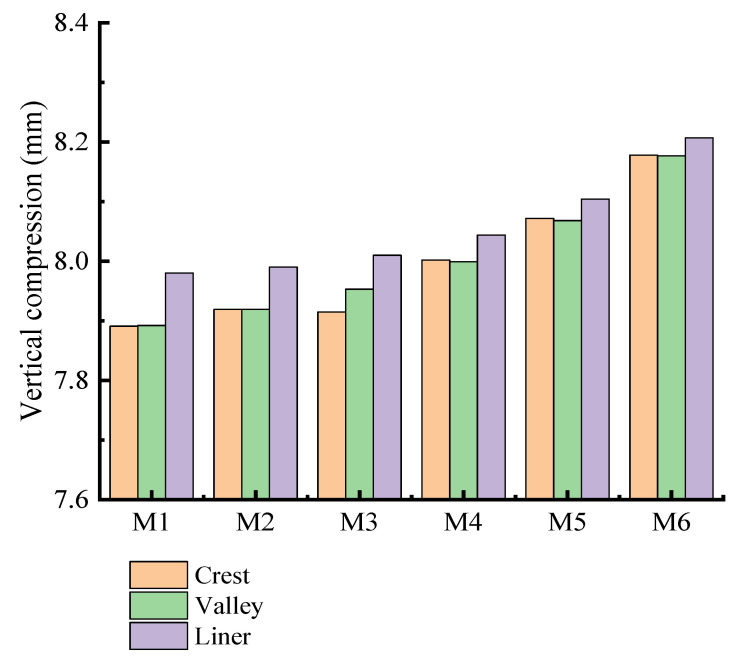
Vertical compression of pipes.

**Figure 17 polymers-14-04000-f017:**
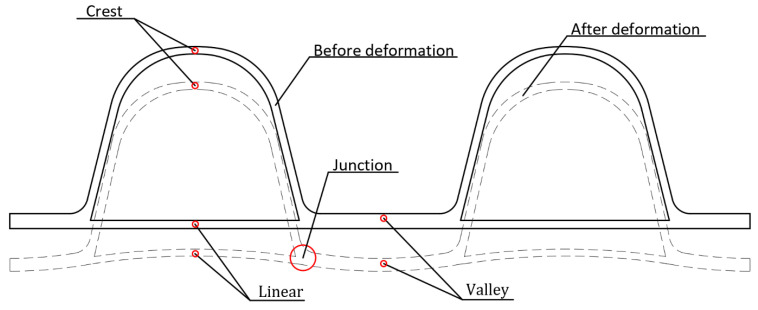
Schematic diagram of cross section deformation.

**Figure 18 polymers-14-04000-f018:**
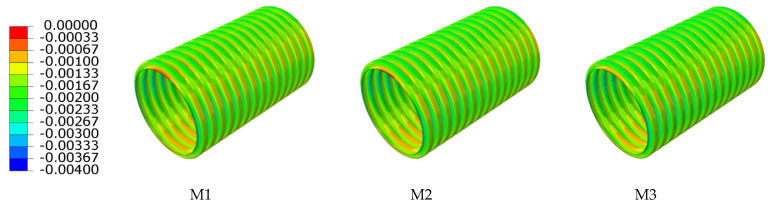
Circumferential strain of exterior wall.

**Figure 19 polymers-14-04000-f019:**
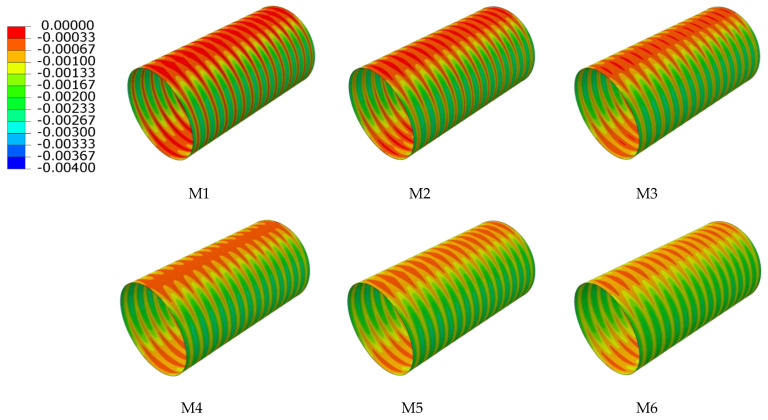
Circumferential strain of interior wall.

**Figure 20 polymers-14-04000-f020:**
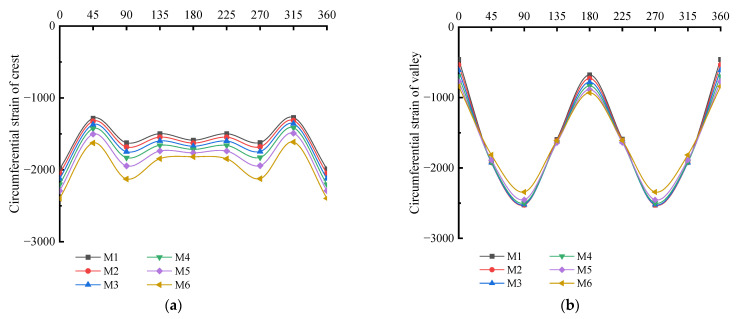
Circumferential strain of various interior and exterior wall thicknesses. (**a**) Crest; (**b**) Valley; (**c**) Liner.

**Figure 21 polymers-14-04000-f021:**
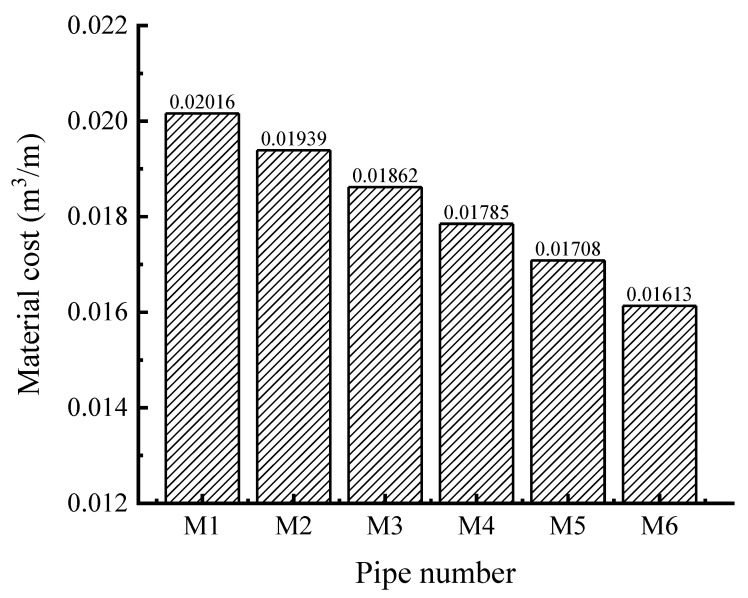
Material cost per unit length of pipe.

**Figure 22 polymers-14-04000-f022:**
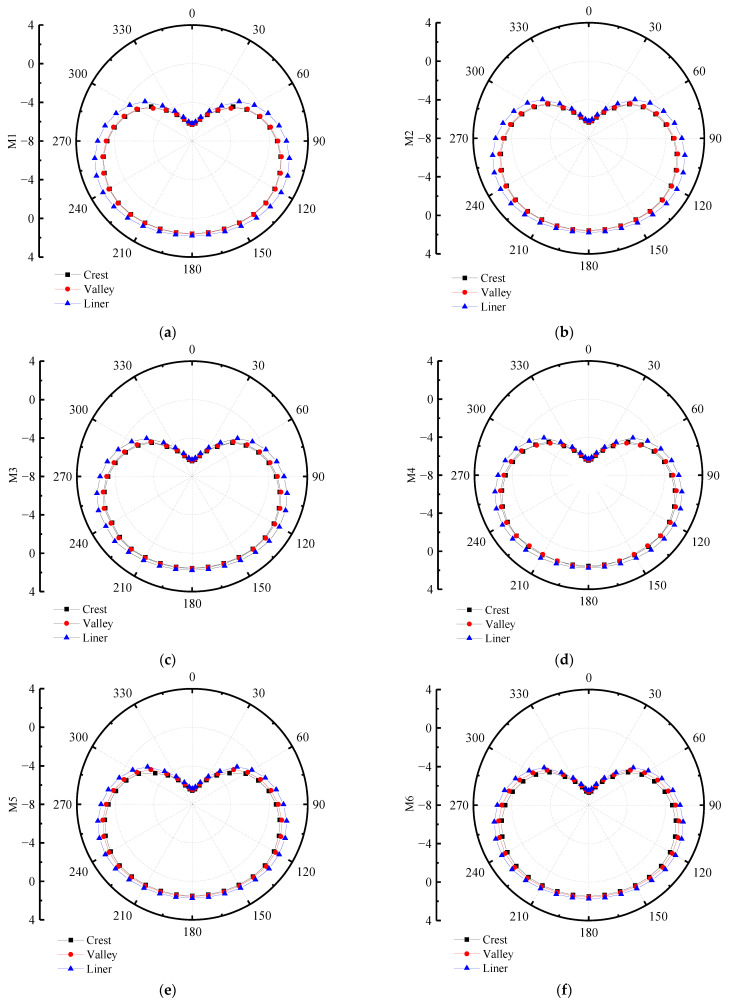
Radial displacement of pipes with various wall thicknesses. (**a**) M1; (**b**) M2; (**c**) M3; (**d**) M4; (**e**) M5; (**f**) M6.

**Table 1 polymers-14-04000-t001:** Material parameter of composites [10].

Blend Composition	Tensile Strength at Yield (Mpa)	Tensile Strength at Break (Mpa)	Young’s Modulus (Mpa)	Elongation at Break (Mpa)
Pure HDPE	22.13 ± 2.23	16.54 ± 1.57		500.00% ± 0%
5%FA masterbatch/95% HDPE	25.60 ± 1.09	19.63 ± 0.01	1495.0 ± 35.4	252.85% ± 56.56%
10%FA masterbatch /90% HDPE	27.55 ± 1.25	21.62 ± 1.01	1477.2 ± 49.9	325.55% ± 51.97%
15%FA masterbatch/85% HDPE	26.06 ± 1.03	20.93 ± 0.60	1370.1 ± 23.2	344.20% ± 22.56%
20%FA masterbatch/80% HDPE	27.13 ± 1.15	22.03 ± 1.47	1327.5 ± 92.3	399.61% ± 40.10%
10%FA masterbatch/5% compatibilizer/85% HDPE	28.82 ± 2.54	23.23 ± 2.80	1451.1 ± 10.8	355.15% ± 9.05%
10%FA masterbatch/10% compatibilizer/80% HDPE	27.22 ± 1.16	22.62 ± 1.01	1360.8 ± 18.8	411.28% ± 51.33%
10%FA masterbatch/15% compatibilizer/75% HDPE	25.56 ± 0.74	21.35 ± 0.36	1297.1 ± 17.1	420.33% ± 61.47%

**Table 2 polymers-14-04000-t002:** Mechanical parameters of each layer of the backfill in the numerical simulation.

Properties	Lay01	Lay02	Lay03	Lay04	Lay05	Lay06	Lay07
Density (kg/m^3^)	1800	1750	1740	1600	1600	1650	1500
Elastic modulus (MPa)	10	15	15	7	9	9	30
Friction angle (°)	35	31	25	28	30	27	30
Cohesive (kPa)	12	10	10	15	15	10	20
Poisson’s ratio	0.3	0.26	0.26	0.23	0.3	0.32	0.35
Compaction degree (%)	90	95	95	90	85	90	

**Table 3 polymers-14-04000-t003:** Properties of pipes’ geometry.

Properties	Value	
Nominal diameter (mm)	800	
Corrugate height (mm)	55	
Corrugate length (mm)	60	
Corrugate spacing (mm)	40	
Pipe number	Interior thickness (mm)	Exterior wall thickness (mm)
M1	2.0	3.0
M2	2.3	2.7
M3	2.6	2.4
M4	2.9	2.1
M5	3.2	1.8
M6	3.5	1.5

**Table 4 polymers-14-04000-t004:** Material parameter of pipes.

Properties	HDPE Corrugated Pipes	FA/HDPE Corrugated Pipes
Density (kg/m^3^)	950	970
Young’s modulus (Mpa)	800	1451.1
Poisson ratio	0.4	0.38

## Data Availability

The data presented in this study are available on request from the corresponding author.

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
