# Peer review of "Research on the Mechanical Behavior of Buried Double-Wall Corrugated Pipes"

_polymers, 2022, doi:10.3390/polym14194000_

Round 1

Reviewer 1 Report

The paper presents an interesting approach based on the Research on the Mechanical Behavior of Buried Double-wall Corrugated Pipes. However, the innovation of the current research work should be further highlighted and emphasized. At the same time, the authors should consider the following comments to greatly improve the quality of the paper.

1. In the abstract, a final statement shall be added that highlights the importance of the study and it's future influence.

2. The introduction needs to be improved by relating to the mechanics of the studied materials and their mechanical characteristics. The references to be included are: 10.1016/j.polymertesting.2017.09.009, 10.1016/j.compstruct.2021.114698, 10.1002/app.46770, 10.3390/polym14132662, 10.1016/j.jiec.2022.06.023, 10.1016/j.porgcoat.2022.107015.

3. The experimental part in section 2 is not detailed. It needs a table showing the raw materials used, manufacturing technology and preparations followed. 

4. The tensile testing shall be reported according to a testing standard. What was the tensile testing speed and how many samples per configuration were examined?

5. Stress-strain curves need to drafted for the five different material compositions, with variance in terms of error bars for each data point. 

6. The tensile test setup, type of grips used and sample geometry of sample tested need to illustrated with dimensions.

7. The formula used for the measurement vertical stress caused by the vehicle load need to referenced. All formulas need to be numbered and referenced.

8. The conclusion needs to be modified to summarize the research outcomes in short statements with clear observations.

Author Response

Dear Reviewer:

Thank you for your letter and for the reviewers’ comments concerning our manuscript entitled “Research on the Mechanical Behavior of Buried Double-wall Corrugated Pipes” (ID: polymers-1905466). Those comments are all valuable and very helpful for revising and improving our paper, as well as the important guiding significance to our researches. We have studied comments carefully and have made correction which we hope meet with approval. Revised portion are marked in red in the paper. The main corrections in the paper and the responds to the reviewer’s comments are as flowing:

Reviewer #1:

  1. Response to comment: In the abstract, a final statement shall be added that highlights the importance of the study it’s future influence.

Response: We are very grateful to the reviewers for pointing out inadequacies in our abstract, at the end of which we briefly describe the highlights of the paper and discuss implications for future research and development.

  1. Response to comment: The introduction needs to be improved by relating to the mechanics of the studied materials and their mechanical characteristics.

Response: It is really true as Reviewer suggested that the introduction in terms of mechanical behavior and material properties is insufficient. We read the six literatures you suggested and found them helpful for our article. All of your suggested articles are now added to the article as important references. The references you suggested are now cited at [1-4,7-8]

  1. Response to comment: The experimental part in section 2 is not detailed. It needs a table showing the raw materials used, manufacturing technology and preparations followed. 

Consider that what you mentioned about polymer preparation and chemical testing is our previous work that has been published. In order to reduce unnecessary disputes and troubles, we do not make it the main content of the article. Therefore, after careful consideration, we hope to focus on the analysis of the mechanical behavior and deformation mechanism of the article, and do not add the preparation and chemical testing of polymers as the main content in the article. In response to your enthusiastic suggestion, we feel very sorry that such a response is made.

If you are interested in the preparation process of FA/HDPE, glad you can read our previous article. The doi: https://doi.org/10.3390/polym13234204.

  1. Response to comment: The tensile testing shall be reported according to a testing standard. What was the tensile testing speed and how many samples per configuration were examined?

We are very sorry for our negligence to provide testing standard, tensile testing spped, and how many samples were checked for each configuration. We have re-written this part according to the Reviewer’s comments and added relevant content to the first paragraph of the second chapter.

  1. Response to comment: Stress-strain curves need to drafted for the five different material compositions, with variance in terms of error bars for each data point. 

Considering the reviewer's suggestion, it was an oversight on our part not to provide the stress-strain curve in the article. We have added the stress-strain curves for different contents of FA/HDPE materials in Figure 4.

  1. Response to comment: The tensile test setup, type of grips used and sample geometry of sample tested need to illustrated with dimensions.

Thanks to the reviewer for directly pointing out our inadequacies, we have selectively added sample size and other content in the second chapter. Due to the failure to pay attention to the grips when doing the tensile test, the accurate grips size information cannot be provided. We are very sorry for not being able to provide this content.

  1. Response to comment: The formula used for the measurement vertical stress caused by the vehicle load need to referenced. All formulas need to be numbered and referenced.

We are very sorry for our negligence of not adding references and numbers to the formulas. We have read your comments carefully and added the reference and number of the formula to the article.

  1. Response to comment: The conclusion needs to be modified to summarize the research outcomes in short statements with clear observations.

We greatly appreciate your pointing out the inadequacies of our conclusions. We analyze and summarize the main research content of the article, and use concise language to explain the research phenomenon and results at the conclusion to make it more prominent and clear.

Special thanks for your review. We did our best to improve the manuscript and made some revisions to it. These changes do not affect the content and frame of the file. Here, we do not list these changes, but they are marked in red in the revised document. We sincerely thank the editors/reviewers for their enthusiastic work and hope that the corrections will be approved. Thanks again for your comments and suggestions

Yours sincerely,

Reviewer 2 Report

This manuscript reports the modellation by finite model effect of corrugated pipes. The experimental parts is clearly written as the entire paper. The paper is ready for publication in Polymers. Only a little thing. I suggest to introduce thermal properties of initial materials

Author Response

Dear Reviewer:

Thank you for your letter and for the reviewers’ comments concerning our manuscript entitled “Research on the Mechanical Behavior of Buried Double-wall Corrugated Pipes” (ID: polymers-1905466). Those comments are all valuable and very helpful for revising and improving our paper, as well as the important guiding significance to our researches. We have studied comments carefully and have made correction which we hope meet with approval. Revised portion are marked in red in the paper. The main corrections in the paper and the responds to the reviewer’s comments are as flowing.

  1. Response to comment: This manuscript reports the modellation by finite model effect of corrugated pipes. The experimental parts is clearly written as the entire paper. The paper is ready for publication in Polymers. Only a little thing. I suggest to introduce thermal properties of initial materials.

Consider that what you mentioned about thermal properties of initial materials is our previous work that has been published. In order to reduce unnecessary disputes and troubles, we do not make it the main content of the article. Therefore, after careful consideration, we hope to focus on the analysis of the mechanical behavior and practical engineering applications of the article, and do not add the thermal properties of polymers as the main content in the article. Hope you can understand. In response to your enthusiastic suggestion, we feel very sorry that such a response is made.

If you are interested in the preparation process of FA/HDPE, glad you can read our previous article,The doi : https://doi.org/10.3390/polym13234204.

Special thanks to you for your good comments.

We tried our best to improve the manuscript and made some changes in the manuscript. These changes will not influence the content and framework of the paper. And here we did not list the changes but marked in red in revised paper.We appreciate for Editors/Reviewers’ warm work earnestly, and hope that the correction will meet with approval.Once again, thank you very much for your comments and suggestions.

Yours sincerely,

Round 2

Reviewer 1 Report

The authors have considered all comments and revised the article professionally. They also were politely responding to each question and seriously acting on each comment to improve their manuscript. Great work. Keep it up.